# Does the Development of Digital Inclusive Finance Promote the Construction of Digital Villages?—An Empirical Study Based on the Chinese Experience

Chengkai Zhang, Yu Li , Lili Yang and Zheng Wang *

Beijing Academy of Science and Technology, Beijing 100089, China; zckzck3317@126.com (C.Z.);
liyu_2016@126.com (Y.L.); yanglili@bjast.ac.cn (L.Y.)
* Correspondence: wangzheng769@126.com

**Abstract:** The degree of the effect of digital inclusive finance on the construction of China's digital villages and the mechanism of action is investigated in this study by matching the digital inclusive finance index in accordance with a data sample of China's provincial digital villages from 2013 to 2020. As indicated by the result of this study, first, the development of digital inclusive finance positively expedites the development of digital villages. Second, geographical and dimensional differences exist when digital inclusive finance boosts the construction of digital villages. Third, digital inclusive finance is capable of facilitating the construction of digital villages by deepening technological innovation and communication infrastructure construction and further enhancing the digital literacy of residents. Fourth, a positive moderating effect of internal conditions of rural residents' consumption and external conditions of financial regulation is reported when digital inclusive finance promotes digital rural development. Based on the above-mentioned findings, the following policy recommendations are presented to advance digital countryside construction in depth. First, following the goal of building Chinese modernization, differentiated policies, with regional resource endowments, social conditions, and rural characteristics considered, should be implemented in accordance with local conditions. Second, the digitalization process in rural areas should be vigorously boosted, and it is imperative to optimize and upgrade mobile communication infrastructure, with the aim of injecting new momentum into China's digital countryside construction. Third, investment in scientific and technological research and development funds and high-level innovative talents should be increased to endow digital technology with better independent innovation capacity and facilitate the level of innovation. Fourth, investment in education should be increased to enhance the digital literacy of urban and rural residents.

**Keywords:** digital inclusive finance; digital village; technological innovation; digital facilities; digital literacy

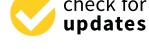


## 1. Introduction

The deep integration of a novel generation of digital technologies (typically big data, artificial intelligence, and cloud computing) with the real economy has formed a new dynamic energy over the past few years for boosting economic and social transformation and expediting industrial structure transformation and high-quality economic growth, which has been extensively discussed worldwide (2023) [1]. The Chinese government stresses the application of digital technology in rural construction and has released the ambitious goal of building China's high-quality digital countryside. Indeed, 2018 Central Government Document No. 1 proposed the implementation of the "Digital Countryside Strategy" for the first time while launching the national digital countryside pilot construction in 2020. Moreover, policy documents (e.g., the Outline of the Digital Countryside Development Strategy and the Action Plan for Digital Countryside Development (2022–2025) [2,3]) have been

released. As of June 2022, the number of Internet users in China's rural areas had reached 293 million, and the rural Internet penetration rate was reported as 58.8%. Furthermore, over 800,000 5G base stations have been opened, such that "5G in counties and broadband in villages" can be achieved. Moreover, digital technology has been extensively used in agricultural production and rural governance, such that a rural digitalization solution can be provided to achieve Chinese modernization. However, some practical obstacles remain that adversely affect the construction of digital villages. On the one hand, under the effects of income and education levels, rural residents in a wide variety of regions have different perceptions and enthusiasm for the digital village, and a significant digital divide has been reported. On the other hand, a wide range of research institutions and technology enterprises have not invested sufficiently in the research and development of digital facilities and equipment, and few interconnected devices have been developed for intelligent collaborative work and automatic digital collection and analysis to provide quality products for the countryside. Accordingly, the introduction of financial capital to facilitate the construction of a digital village, to narrow the "digital divide" among residents, and to expedite enterprises to serve villagers, has attracted widespread attention from society. China urgently needs to explore mechanisms and pathways for digital inclusive finance to boost the construction of digital villages. Under such circumstances, this study has strong practical significance and policy guidance value.

This paper attempts to conduct an in-depth study from two perspectives. First, this paper explores the impact of digital financial inclusion on digital village construction. Secondly, we explore the paths, heterogeneity, and role played by special factors that digital inclusive finance will produce in the process of influencing the construction of digital villages. Thus, the marginal contributions of this study are elucidated as follows. First, the driving effect of digital inclusive finance on the digital countryside and the path mechanisms underlying the existence of technological innovation, communication infrastructure, and residents' education level is explained theoretically and systematically, and a digital countryside development degree index is built following the panel data on 30 Chinese provinces and cities from 2013 to 2020. Second, a panel model is used to verify that digital inclusive finance has a positive and significant impact on the digital countryside. Again, a mediating effects model is used to test the hypotheses derived from the theoretical analysis. Lastly, a moderating effect model is used to analyze the moderating effect of the intrinsic basis of consumption of rural residents and the external conditions of financial regulation on the existence of digital inclusive finance affecting the digital countryside.

The remainder of this paper is organized as follows: Section 2 presents a literature review and the research hypotheses. Section 3 presents the data sources and the main methodology. Section 4 presents the empirical results. Section 5 presents the conclusions and recommendations.

## 2. Literature Review and Research Hypothesis

### 2.1. Literature Review

Scholars have focused on the concept and measurement of digital inclusive finance to facilitate the development of the "three rural areas" and to expedite the construction of a digital village, among other perspectives. First, the concept and measurement of digital inclusive finance was first introduced by the United Nations in 2005, with the aim of preventing the emergence of financial exclusion. Inclusive finance aims at offering comprehensive and accessible financial services to a wide variety of groups in society (1993) [4] and is more universal and accessible than conventional finance. Most of the existing research regarding inclusive finance has been analyzed for the utilization rate, accessibility, and development of inclusive finance (2011) [5]. Nevertheless, in the process of development, some financial institutions do not support the sustainable development of inclusive finance programs due to high transaction costs and high risks. With the development of technologies (e.g., big data, artificial intelligence, and 5G), inclusive finance and digital technology have been integrated with each other, the development of inclusive

finance has entered the stage of information technology, and digital inclusive finance has come into being. In accordance with the G20 Advanced Principles for Digital Inclusive Finance released by the People's Bank of China in 2016, digital inclusive finance refers to inclusive finance achieved by relying on digital technology, thus making financial services more accessible. Digital inclusive finance refers to a multi-dimensional concept; therefore, when researching digital inclusive finance, the relevant indexes of finance and digital technology should be considered to accurately indicate its degree and level of development. The Digital Inclusive Finance Index (2011–2020) (2020) [6] report released by a group of Peking University's Digital Finance Research Centre is representative of a considerable number of research results. Its indexes were selected to consider three dimensions of digital inclusive finance, i.e., breadth of coverage, depth of use, and degree of digital support services. To be specific, a total of 33 sub-indexes were selected for portrayal, and data samples from China's provinces, cities, and counties from 2011 to 2020 were measured to determine the credible indexes for the level of development of digital inclusive finance. Although digital inclusive finance has existed for a short period, it has become a focal issue for scholars, especially the issue regarding the degree of development of digital inclusive finance. Although there are some differences in perspectives among studies, there is agreement on the construction of the specific measurement system and the selection of indexes, that is, digital inclusive finance is multi-dimensional and dynamic in nature.

Second, research has focused on using digital inclusive finance to expedite the development of the "three rural areas". To facilitate the development of the "three rural areas", scholars have mostly conducted research from two perspectives: macro and micro. On the one hand, from a macro perspective, scholars have investigated how to use digital inclusive finance to boost the revitalization of China's rural areas. At the new stage of consolidating and expanding the achievements of poverty eradication, scholars suggested that digital inclusive finance will conform to the laws of rural industrial development, provide effective digital financial support for rural revitalization while ultimately ensuring that the rural revitalization strategy is implemented effectively (2022) [7]. Some scholars suggested that the weakness of the rural financial system is the main obstacle to the development of rural industries, and the development of digital inclusive finance has notably compensated and facilitated the exclusion phenomenon of rural finance, promoted the gradual advancement of rural industries, and consolidated the achievements of rural revitalization (2022) [8]. From the perspective of empirical evidence, scholars suggested that digital inclusive finance is capable of exerting a mechanized universal effect, entrepreneurial incentive effect, and income growth effect to facilitate the development of rural revitalization (2023) [9]. On the other hand, the micro perspective explores how digital inclusive finance can be used to expedite the development of the agricultural industry, residents' income and consumption, and residents' innovation and entrepreneurship. Most scholars have a positive view on the use of digital inclusive finance for facilitating the integration of rural industries (2021; 2023; 2023) [10–12], suggesting that digital inclusive finance is capable of addressing the financing constraints of farmers, expanding the convenience and accessibility of financial services for rural residents, and positively expediting agricultural industrialization. Scholars argued that deep theoretical logic is required for digital inclusive finance to help farmers increase their incomes, such that incomes can be generated by increasing farmers' wages, business, transfer, and property incomes to achieve the goal of common prosperity (2022) [13]. It has also been suggested that enhancing rural consumption has a significant positive effect on boosting the rural economy, and digital inclusive finance can rely on digital information redemption to innovate financial services and enhance financial accessibility, thus stimulating rural consumption growth (2021) [14]. Scholars have different views on the capability of digital inclusive finance to facilitate innovation and enhance entrepreneurship among farmers. It was argued that digital inclusive finance development can not only increase the availability of credit for farmers to access entrepreneurial finance (2021) [15] but also reduce transaction costs and enhance financial capability by promoting the use of electronic payments and the use of internet media, thereby promoting their entrepreneurship. Al-

though digital inclusive finance promotes farmers' investment in production and business, the effect of digital inclusive finance on farmers' investment in agricultural production and business is not significant. It only promotes farmers' investment in industrial and commercial production and business (2023) [16].

Lastly, there are aspects of research on the use of digital inclusive finance to facilitate the digital countryside. The construction of a digital countryside is a strategic direction and an important method for rural revitalization, and thus it is an important element in building digital China (2019) [17]. Digital inclusive finance is capable of promoting data integration, enhancing residents' digital financial literacy, creating novel financial scenarios, and reducing financial transaction costs. Thus, it is conducive to consolidating the information technology infrastructure and promoting the digital countryside as a rural solution for Chinese-style modernization (2022) [18].

However, it is noted that the earlier literature placed a greater focus on the effect of digital inclusive finance on industrial structure, residents' income and consumption, and the digital countryside itself. Few studies considered the two as an organic whole and investigated the effect of digital inclusive finance on the construction of the digital countryside and the possible driving mechanisms. Thus, in this study, the theoretical role of digital inclusive finance in driving the digital countryside and the existence of path mechanisms underlying technological innovation, communication infrastructure, and residents' education level are first systematically explained, and then a digital countryside development degree index is set using panel data on 30 Chinese provinces and cities from 2013 to 2020. The hypotheses derived from the theoretical analysis are tested using a mediating effects model with the aim of further exploring possible regional and dimensional variability. Compared with the existing research, the above research components constitute possible marginal contributions.

*2.2. Research Hypothesis*

From a theoretical point of view, the use of digital inclusive finance to solve social development problems is an important frontier direction for financial development theory in development economics, which was initially designed to solve the problem of financial exclusion, and gradually developed to financial inclusion and even financial equity. As early as the 1960s, scholars at the structuralist school (1955) [19] proposed that the practice of focusing on capital formation and resource allocation in developing countries would lead to backwardness in the financial structure and would constrain the realization of financial inclusion, with the variety and number of financial institutions and financial instruments stagnating at a low level. During the 1980s and 1990s, the emergence of the new Keynesian and endogenous growth theories (2012) [20] illustrated that restricting banking access and direct financing could exclude financial service objects and service agents, and the inclusive behavior of financial services was limited by policy orientation. From the end of the twentieth century to the present day, neo-liberalism and new institutional economics (1991; 2000) [21,22] reconsidered financial inclusion, arguing that poor countries and poor communities cannot afford to cover the costs of financial system formation, and thus financial exclusion remains. Therefore, government subsidies are needed to reduce the costs of financial inclusion and to protect the population in disadvantaged areas.

From a practical point of view, digital financial inclusion is a new concept arising at the intersection of the fields of finance and technology. At its core, digital financial inclusion uses technology as the first element to drive the financial industry to expand innovative service tools, which, as a supplement to traditional finance, can expand the coverage of financial services and provide more residents and enterprises with the financial products they need. The comprehensive rise of digital inclusive finance has expedited the financialization of the ecological map of the technology industry, revolutionized the development pattern in the conventional financial industry, broadened the technological path of financial services for the three rural areas, and provided novel ideas for farmers to increase their income while boosting enterprise development and rural construction (2021) [23]. First, digital

inclusive finance has expanded the reach and coverage of financial services using advanced digital technologies (e.g., cloud computing and big data). With a novel Internet platform built and the threshold of financial business availability lowered, financial services have been offered to more rural long-tail people, thus circumventing the financial discrimination that exists in conventional institutions. Digital inclusive finance has allowed rural residents to obtain the funds required for production and living at low cost, thus improving their quality of life, broadening their business undertakings, and fundamentally enhancing the demand for digital village construction. Second, digital inclusive finance can be leveraged with innovative technologies (e.g., interactive artificial intelligence and complex spectrum blockchain) to integrate internet data for agriculture-associated enterprises and MSMEs instead of conventional technical paths, thus enhancing the digitalization of enterprises, ensuring the healthy development of agriculture-associated enterprises and MSMEs with the cooperation and support of considerable capital, and boosting the growth of enterprises and the construction of villages simultaneously. Lastly, complemented by the central government's inclusive finance policy for the "three rural areas", digital inclusive finance can cooperate with township governments in rural construction and provide targeted financial services. With the support of policies and funds, resources (e.g., population, technology, land, and other digital rural construction elements) will rapidly converge to facilitate the digitalization of townships for further breakthroughs in information infrastructure, rural government governance, intelligent agricultural production, and other aspects such that a comprehensive digital rural construction can be achieved.

Based on the above logical analysis, research Hypothesis 1 is proposed in this study.

**H1.** *Advances in digital inclusion can positively contribute to building a digital village.*

In the process of exploring digital inclusion to facilitate the construction of digital villages, scholars have found that there are important pathways that influence the correlation between the two, as shown in Figure 1. Technological innovation, infrastructure, and digital literacy are the key directions that scholars have focused on.

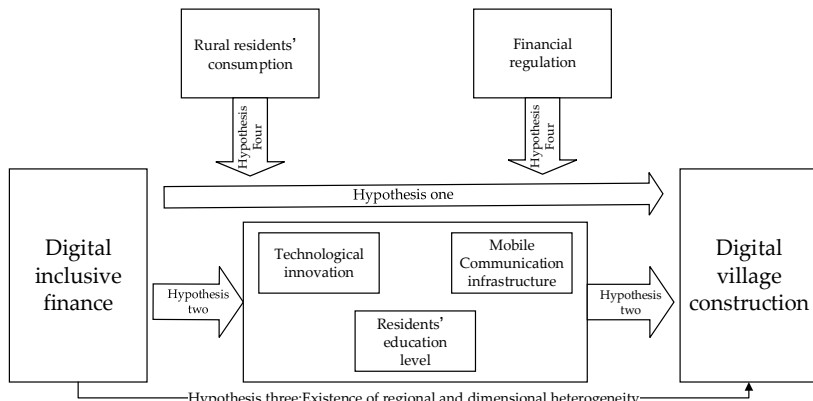

**Figure 1.** The mechanism underlying the mediating effects of digital inclusive finance on promoting digital village building.

The exploration and development path of technological innovation cannot be achieved without the fundamental role played by finance (2003) [24]. With the advances in digital inclusive finance, diverse digital financing channels have become a novel option for firms. As revealed by existing research, access to funds for technological innovation can be constrained by information asymmetry (2017) [25]. A central point of boosting regional development refers to facilitating a smooth and efficient flow of innovation funds to innovative technology firms (2007) [26] so that firms are enabled to create more social wealth. Digital inclusive finance can process big data information at low cost and low risk (2018) [27] while reducing the information asymmetry problem between credit parties using digital technology tools, activating the matching between resource elements and

rural construction projects, enhancing the activity of enterprises' technological innovation activities, and addressing the financial difficulties of enterprises' technological innovation. To access the rural market and maximize their profits, enterprises with sufficient funds will further enhance their technological development to improve their production efficiency and provide digital products that are more relevant to the rural market. As a result, digital governance products based on innovative technologies (e.g., industrial big data, interactive artificial intelligence, and regional digital twins) have emerged. Innovative enterprises can technically support the construction of digital villages, and their cooperation model with grassroots governments has forged a novel map for constructing regional digital villages.

The breadth of mobile communication infrastructure coverage markedly affects the progress of digital inclusive finance (2007) [28]. Driven by profit, financial commercial institutions are inclined to use the technical advantages of modern information (e.g., the Internet and big data), with the aim of addressing the limitations of financial services in time and space, expanding the rural user profile, widening the scope of user services, and achieving profit growth. Financial and commercial institutions have joined with enterprises and local governments to further invest in the construction of mobile communication infrastructure following the guidance of the rural inclusive finance policy, with the planned construction of long-distance fiber optic cable lines, microwave base stations, data processing centers, and other hardware conditions in vast rural areas. Moreover, the construction of mobile communication infrastructure will drive the development of digital industries in rural areas, stimulate the rapid emergence of rural e-commerce industries, achieve a win-win situation in terms of increasing farmers' income and improving the appearance of the countryside, and provide hardware guarantees for the construction of digital villages.

An improvement in residents' digital literacy is significantly correlated with the development of regional digital finance, and the emergence of digitally inclusive finance will further bridge the digital divide, with the aim of enhancing the digital literacy of urban and rural residents. Scholars suggested that digital inclusive finance is capable of addressing the problem of financial cost inversion with accurate publicity, credit, services, and risk identification, thus facilitating the equalization of financial services for urban and rural residents and enhancing the digital literacy of residents to bridge the digital divide. A higher digital literacy of residents has a catalytic effect on the construction of the digital countryside, such that the conventional agricultural production and living model will be broken, and new industries and models for agricultural production, agricultural product processing and circulation, and agricultural leisure tourism with regional characteristics will be created under the e-commerce platform, a live streaming platform and tourism platform, and a new ecology of the rural digital economy will be built.

Based on the above logical analysis, this study proposes research Hypothesis 2.

**H2.** *Digital inclusion can be promoted in the digital village by enhancing technological innovation, communication infrastructure development, and digital literacy of the population.*

Given the differences in the degree of development of digital inclusive finance that arise from the economic and regional rural development gap in the eastern, central, and western areas of China, a differentiation analysis should be conducted. First, the geographical differences should be analyzed. Significant regional differences exist in the total economic development and regional rural construction in China. In particular, the total economic share decreases in the eastern, central, and western areas, in that order, whereas the scale of rural areas increases in the eastern, central, and western areas, in that order (2022) [29]. There is also a regional concentration of financial resources (2016) [30], with eastern cities ranking first for the excessive concentration of financial resources, while western cities rank at the bottom for the level of financial resource aggregation (2016) [31]. Financial exclusion is more serious in remote areas of central and western China. Second, the dimensional differences should be analyzed. In this study, digital financial inclusion indexes are selected to cover three first-level indexes, i.e., breadth of coverage, depth of use,

and degree of digital support services (2020) [32]. To be specific, the breadth of coverage uses the number of card-tied users, the number of accounts, and the proportion of card-tied users of Alipay as the secondary indexes to show the degree of account coverage; the depth of use uses credit, payment, and credit business as secondary indexes to indicate the effect of digital inclusive finance development; the degree of digital support services uses mobile, affordable, and convenience as secondary indexes to indicate the maturity of Internet technology.

Based on the above logical analysis, Hypothesis 3 is proposed.

**H3.** *Geographical variability and dimensional differences exist in the process of digital inclusive finance for the digital village (Figure 1).*

The idea that the growth in consumption of rural residents will stimulate the further development of digital inclusive finance, give rise to the diversity of financial derivatives, increase the activity of capital in rural areas, and be the driving force behind the intrinsic development of inclusive finance is well recognized by scholars (2022) [33]. However, the extent to which this growth in consumption will affect the development of digital inclusive finance to promote the development of the digital countryside is an important issue that we need to discuss at present. Similarly, previous research on financial regulation in digital inclusive finance to promote the development of digital villages has led to the publication of valuable views by scholars. They believe that (2019) [34] strengthening financial regulation can effectively reduce liquidity risk and the emergence of illegal arbitrage problems, thus regulating the financial market so that those who have a real need can enjoy the power of development through the channels of digital inclusive finance.

**H4.** *In the process of using digital inclusive finance to promote the construction of digital villages, residents' consumption, and financial regulation play a moderating role.*

### 3. Research Design

*3.1. Selection and Description of Variables*

3.1.1. Explanatory Variables

Digital village (DR). In order to measure the level of digital village construction more comprehensively, systematically, and objectively, this paper uses the digital village evaluation system constructed by scholars in previous studies (2023) [35] to analyze the system. This involves a total of four dimensions: financial investment, infrastructure, agricultural production, and living services, and uses the entropy weighting method to measure the degree of digital village development in each province and city, with index results ranging from 0 to 1. A scatter plot showing the relationship between digital inclusive finance and the digital village development index is shown in Figure 2.

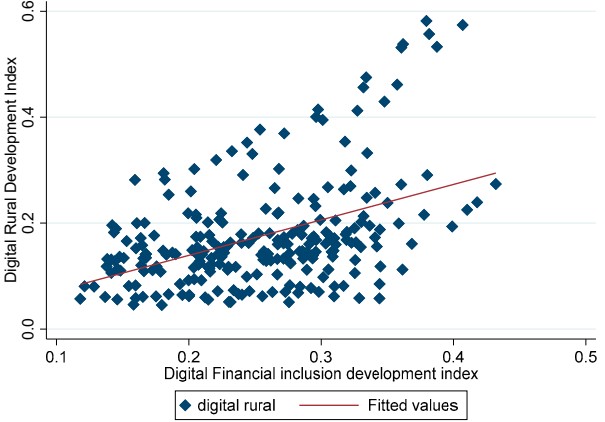

**Figure 2.** A Scatterplot Showing the Relationship between Digital Inclusive Finance and the Digital Village Development Index.

### 3.1.2. Core Variables

Digital inclusive finance (DIF). Digital inclusive finance is measured using the China Digital Finance Development Index published by Peking University's Digital Finance Research Centre. This index system covers three dimensions: breadth of digital finance coverage (DIF1), depth of use (DIF2), and degree of digitalization (DIF 3), and this study spans the period 2013–2020. This index is also widely used in current practice to examine the level of development of digital financial inclusion. The distribution of the digital financial inclusion index is modeled using the Epanechnikov kernel function and is shown in Figure 3.

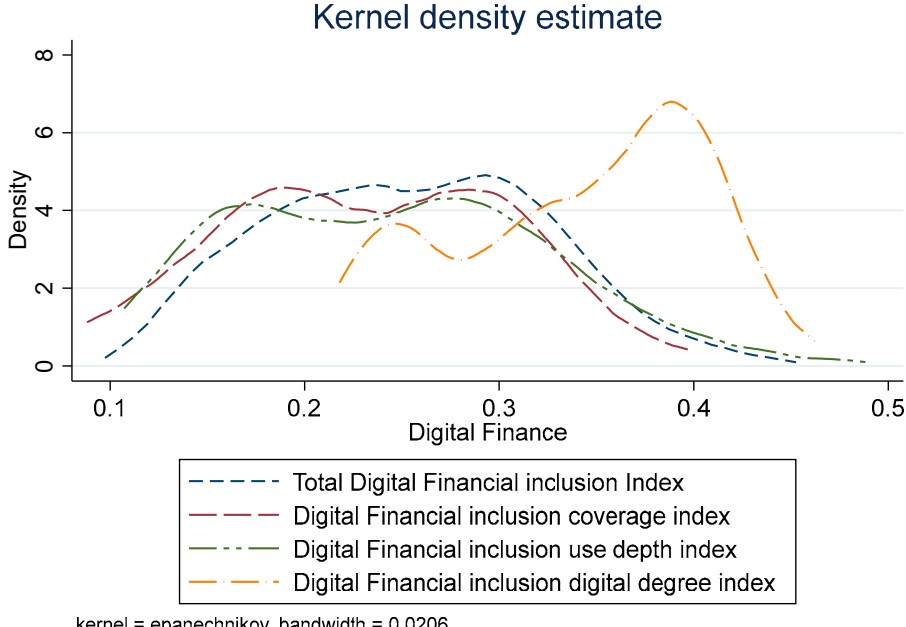

**Figure 3.** Digital Inclusive Finance Kernel Density Distribution Map.

### 3.1.3. Mediated Transmission Variables

The degree of science and technology innovation (TI) [36]. Science and technology innovation is an important driving force for the construction of digital villages, which can effectively improve the landscape of villages. The number of patent applications granted is an important index for the degree of science and technology innovation. Accordingly, we use the number of patent applications granted in a respective province to represent the level of science and technology innovation in that province.

Mobile communication infrastructure (MCI) [37]. Mobile communication infrastructure is an important aspect in the development of digital inclusion finance, and the level of its construction has a profound impact on the degree of digital inclusion coverage. Thus, we use mobile phone exchange capacity to examine the level of mobile communication infrastructure construction in a respective province.

Digital literacy of residents (DL) [38]. Residents can improve their digital literacy by receiving external digital training, and higher digital literacy will promote digital financial inclusion. Therefore, we use the number of full-time information and digitalization teachers in general higher education institutions to measure the education level of residents in a respective province, with a higher number representing a higher emphasis on the improvement of residents' digital literacy in that region.

### 3.1.4. Control Variables

Level of economic development (EDI) [39]. The level of economic development can positively contribute to the construction of digital villages, which is measured in this study using the per capita gross regional product of the respective province.

Industrial structure status (IS) [40]. The optimization of industrial structures will directly affect the allocation of resources, and the optimization of industrial structures will facilitate the construction of digital villages. In this study, we use the value added of the secondary industry as a proportion of the tertiary industry to express industrial structure status.

Socio-population density (SPD) [41,42]. The greater the population density of a region, the greater the demand for digital village construction. In this study, we use the resident population of a region as a proportion of land area to examine the level of population density.

Financial activity of residents (FAR) [43]. The level of resident financial activity is positively related to the acceptance of digital inclusive finance, implying that a region is more capable of attracting capital, which can alleviate the funding problem of digital village construction. In this study, the sum of resident savings and loans is expressed as a proportion of GDP.

Digital infrastructure level (DIL) [44]. The promotion of digital infrastructure creates a fundamental development environment for the construction of digital villages and is an important part of the effect of digital construction. In this study, the digital infrastructure level is represented by the length of fiber optic cable lines.

*3.2. Model Setting*

3.2.1. Baseline Model

$$Y_{it} = a_0 + a_1 DIF_{it} + a_2 control_{it} + \xi_i + \phi_i + \varepsilon_{it} \tag{1}$$

where the explanatory variable $Y_{it}$ denotes digital village construction, the core explanatory variable, $FT_{it}$ is digital financial inclusion, and $control_{it}$ is the control variable. $\xi_i$ represents the area fixed effects, $\phi_i$ is a time fixed effect, and $\varepsilon_{it}$ is the error term.

3.2.2. Mediating Effects Model

The mediating effects model proposed by Baron (1986) [45] and other scholars was used to study the mechanism underlying the effect of digital inclusive finance on the digital village. When measuring the effect of the independent variable $X$ on the dependent variable $Y$, if $X$, by influencing the variable $M$, has an effect on $Y$, it is called a mediating variable.

$$Y = cX + e_1 \tag{2}$$

$$M = aX + e_2 \tag{3}$$

$$Y = c'X + bM + e_3 \tag{4}$$

$$c = c' + ab \tag{5}$$

The above regression equation is used to describe the correlation between the variables. The coefficient of Equation (2) $c$ is the total effect of the independent variable $X$ on the dependent variable $Y$, the total effect of the dependent variable. The coefficient of Equation (3) $a$ is the total effect of the independent variable $X$ on the mediating variable $M$ of the dependent variable. The coefficient of Equation (4) $b$ denotes the effect of the independent variable on the mediating variable after controlling for the effect of the mediating variable $M$ on the dependent variable $Y$. The coefficient $c'$ is the effect of the mediating variable on the dependent variable after controlling for the effect of the mediating variable $M$. After controlling for the effect of the mediating variable, the direct effect of the independent variable $X$ on the dependent variable $Y$ is determined. $e_1$, $e_2$, and $e_3$ represent the regression residuals.

### 3.2.3. Coordinated Effects Model

To further examine the effect of other factors on the contribution of digital inclusive finance to the development of the digital village, interaction terms are introduced to investigate the effects of the moderating variables, as written in the following:

$$Y_{it} = \beta_1 + \beta_2 DIF_{it} + \beta_3 ADJUSTER_{it} + \beta_4 DIF_{it} \times ADJUSTER_{it} + \beta_5 control_{it} + \epsilon_{it} \quad (6)$$

where $Y_{it}$ denotes the construction of digital villages in province i in year t; $DIF_{it}$ represents the development of digital inclusive finance in province i in year t; and $ADJUSTER_{it}$ expresses the condition of the moderating variable in province i in year t. The interaction term $DIF_{it} \times ADJUSTER_{it}$ represents the effect of the moderating variable on the effect of digital inclusive finance on digital villages. $control_{it}$ is the control variable of the model, $\beta_{1-5}$ represents the constant term, and $\epsilon_{it}$ expresses the random error term.

### 3.3. Data Sources and Descriptive Statistics of Variables

In this paper, the research sample is provincial digital financial inclusion and digital village panel data with a data retrieval date of April 2023. StataMP17 (64-bit) software is used to measure and analyze the panel data. The digital financial inclusion index is from the Digital Financial Inclusion Index compiled by the Centre for Digital Finance at Peking University, and the digital countryside measures and other macro variables are from the China Statistical Yearbook, China Rural Statistical Yearbook, China Urban Statistical Yearbook, and the Ministry of Agriculture and Rural Development of the People's Republic of China, etc. This study spans the period from 2013 to 2020. Furthermore, for data processing, variables (e.g., the Digital Inclusive Finance Index) are determined by dividing the original value by 100, and several missing values for indexes are estimated using interpolation. Table 1 lists the descriptive statistics for the specific variables.

**Table 1.** Descriptive statistics of variables.

| Variable Category | Variable Name | Variable Symbol | Average Value | Standard Deviation | Maximum Value | Minimum Value |
|---|---|---|---|---|---|---|
| Explained variables | Digital village | DR | 0.149 | 0.103 | 0.582 | 0.045 |
| Explanatory variables | Digital inclusive finance | DIF | 0.256 | 0.069 | 0.432 | 0.118 |
| | Digital inclusive finance—breadth of coverage | DIF1 | 0.233 | 0.072 | 0.397 | 0.088 |
| | Digital inclusive finance—depth of use | DIF2 | 0.247 | 0.079 | 0.489 | 0.107 |
| | Digital inclusive finance—the degree of digitization | DIF3 | 0.354 | 0.064 | 0.462 | 0.218 |
| Intermediate variables | Science and technology innovation | TI | 3.336 | 9.530 | 70.972 | 0.050 |
| | Infrastructure | MCI | 6.499 | 4.953 | 23.804 | 0.927 |
| | Education level | DL | 4.705 | 3.012 | 13.340 | 0.380 |
| Control variables | Level of economic development | EDI | 4.946 | 2.761 | 16.416 | 2.209 |
| | Regional industrial structure | IS | 4.538 | 15.626 | 98.863 | 0.944 |
| | Population density | SPD | 2.929 | 7.151 | 39.492 | 0.079 |
| | Resident financial activity | FAR | 3.169 | 1.148 | 8.131 | 1.664 |
| | Digital infrastructure level | DIL | 0.912 | 0.827 | 3.990 | 0.074 |

## 4. Empirical Study

### 4.1. Correlation Analysis

Table 2 lists the results of the variable correlation tests. As indicated by the preliminary results, digital villages develop a strong positive correlation with digital inclusive finance (DIF) and the economic development level (EDL), regional industrial structure (IS), population density (PD), and level of transmission infrastructure (LOC), whereas the effect on the financial activity of the population (FA) does not take on any significance. The

sign and significance of the regression coefficients vary with the control of the variables, suggesting that the preliminary analysis still requires in-depth validation using regression. As indicated by the additional VIF test results, the value of the VIF variance inflation factor among the variables is 7.57, and being less than 10, this suggests that the multicollinearity among the variables in this study's empirical model is relatively poor and can be subjected to the next step of regression analysis.

**Table 2.** Correlation analysis.

| Variables | DR | DIF | EDL | IS | PD | FA | LOC |
|---|---|---|---|---|---|---|---|
| DR | 1.000 | | | | | | |
| DIF | 0.444 *** | 1.000 | | | | | |
| EDL | 0.437 *** | 0.653 *** | 1.000 | | | | |
| IS | 0.093 *** | 0.323 *** | 0.764 *** | 1.000 | | | |
| PD | 0.201 *** | 0.288 *** | 0.713 *** | 0.948 *** | 1.000 | | |
| FA | −0.039 | 0.391 *** | 0.583 *** | 0.636 *** | 0.517 *** | 1.000 | |
| LOC | 0.842 *** | 0.515 *** | 0.227 *** | −0.159 ** | −0.047 | −0.179 ** | 1.000 |

"***" and "**" indicate significance at the 1% and 5% levels of significance.

### 4.2. Baseline Regression Results

Since the data used are continuous variables, an OLS model was used to carry out the regression estimation of the effect of digital inclusive finance on digital village construction. The *p*-value of Hausman's test was less than 0.01, and the original hypothesis was rejected, thus a two-way fixed effects model was used. Table 3 lists the test results for the effect of digital inclusive finance on digital village construction from model (1). In the baseline regression, this study uses a progressive regression strategy, with column (1) adding time and individual fixed utility but not any control variables. This method shows that the coefficient of digital inclusion finance passes the significance test at 1%. Columns (2)–(6) progressively add control variables, and the coefficient for digital inclusion decreases over the course of adding further variables but remains positively significant at 1%. With model (6) as the benchmark regression result, in an economic sense, for every 1% increase in the digital inclusive finance index, the construction of digital villages will be increased by 1.037, i.e., digital inclusive finance is capable of notably boosting the construction of digital villages, and a significant positive correlation is reported between the two. Thus, Hypothesis 1 holds.

Regarding the control variables, the level of economic development has a significant positive effect on the digital countryside, thanks to China's long-term rapid economic growth and rising GDP per inhabitant, which has driven the development of the countryside. Changes in the industrial structure have a significant negative impact on the construction of the digital countryside. The core of the development of the digital countryside still needs to focus on agricultural production and raising the income level of farmers to drive the development of the countryside. The prosperity of the secondary industry has less impact on the construction of the countryside, so this may be one of the important reasons why changes in the industrial development structure have a significant negative impact on the construction of the digital countryside. Furthermore, population density exerts a significantly positive effect on the construction of the digital countryside. Areas exhibiting high population density raise a more prominent demand for the construction of the digital countryside, and the new generation of rural residents yearns to cut away from the conventional countryside scene at the levels of production and operation, education and healthcare, and cultural exchange, and look forward to integrating into the new modern, informative, and digital life. There is a positive contribution of residents' financial activity to the construction of the digital countryside. The higher the level of residents' financial activity, the higher the residents' living standard will be indicated laterally, and the higher the residents' expectations for the construction of the digital countryside will be. Digital infrastructure construction can significantly promote the construction of China's digital

countryside. Over the years, China's construction of digital facilities has directly contributed to the development of the countryside while indirectly increasing the productivity of the primary industry.

**Table 3.** Results from the empirical test of the efficiency of digital inclusive finance on financial support for digital village construction.

| Variables | (1) DR | (2) DR | (3) DR | (4) DR | (5) DR | (6) DR |
|---|---|---|---|---|---|---|
| DIF | 2.607 *** (0.314) | 2.051 *** (0.387) | 1.896 *** (0.360) | 1.110 *** (0.291) | 1.197 *** (0.282) | 1.037 *** (0.242) |
| EDI | | 0.012 ** (0.005) | 0.037 *** (0.006) | 0.028 *** (0.005) | 0.033 *** (0.005) | 0.030 *** (0.004) |
| IS | | | −0.005 *** (0.001) | −0.005 *** (0.001) | −0.006 *** (0.001) | −0.004 *** (0.001) |
| SPD | | | | 0.220 *** (0.019) | 0.238 *** (0.019) | 0.190 *** (0.018) |
| FAR | | | | | 0.022 *** (0.006) | 0.022 *** (0.005) |
| DIL | | | | | | 0.048 *** (0.006) |
| CONSTANT | −0.422 *** (0.072) | −0.438 *** (0.071) | −0.277 *** (0.072) | −8.568 *** (0.734) | −9.429 *** (0.747) | −7.623 *** (0.676) |
| Fixed time | Yes | Yes | Yes | Yes | Yes | Yes |
| Fixed area | Yes | Yes | Yes | Yes | Yes | Yes |
| R2 | 0.926 | 0.928 | 0.939 | 0.962 | 0.965 | 0.9744 |

"***" and "**" indicate significance at the 1% and 5% levels of significance.

### 4.3. Stability Test and Endogeneity Treatment

To ensure the reliability of this paper's conclusions, this section tests for possible endogeneity and stability issues, the results of which are listed in the Table 4.

**Table 4.** Robustness test results.

| Variables | DR (Excluding Geographical Panel Data) (1) | DR (Excluding Time Panel Data) (2) | DR (Replaces Core Variables) (3) |
|---|---|---|---|
| FT | 0.738 *** (0.253) | 1.383 *** (0.396) | |
| FTI | | | 0.597 *** (0.70) |
| Control | Yes | Yes | Yes |
| Constant | −7.735 *** (0.836) | −7.571 *** (1.191) | −4.434 *** (0.727) |
| Fixed time | Yes | Yes | Yes |
| Fixed area | Yes | Yes | Yes |
| R2 | 0.975 | 0.983 | 0.980 |

"***" indicates significance at the 1% level of significance.

4.3.1. Regression Tests for Excluded Data Samples

To further test the stability of the effect of digital inclusive finance development on the construction of digital villages, the following two methods are used in this study. One is to exclude geographical panel data, with samples randomly excluded from five provinces and cities including Beijing, Jilin, Fujian, Guangdong, and Yunnan, and a two-way stationary regression test is performed. The regression results are listed in column (1) of Table 4, suggesting that the regression coefficient of 0.738 for the randomly excluded provinces remains significant at 1%. Thus, the results of this study are robust. Second, after excluding

the time panel data, the development of digital inclusive finance shows a close correlation with the national top-level design. Thus, the introduction of macro policies will effectively boost the formation of new ecology, novel models, and new tools for digital inclusive finance, such that the digital countryside will be empowered to gain. In the sample time series of this study, there is an exogenous policy shock, i.e., the State Council's release of the "Plan for Promoting the Development of Inclusive Finance (2016–2020)". Accordingly, this study excludes the sample data before 2013 (inclusive) and conducts regression analysis on the remaining panel data, as listed in column (2) of Table 4. As revealed by these results, the regression coefficient of 1.383 is significant at 1%, which is consistent with the overall regression results, and thus the robustness of this study is reliable.

### 4.3.2. Replacement of Core Explanatory Variables

In this study, a comprehensive evaluation system of digital inclusive finance is developed for 30 provinces (with Tibet, Hong Kong, Macau, and Taiwan excluded) from 2013 to 2020 in the perspectives of both inclusiveness and digitality. Subsequently, the entropy value method is applied so that the level of digital inclusive financial development (FTI) measured with this evaluation system can be used as a replacement variable for the core explanatory variables in the model. The dimensional design and specific indexes are defined as follows. Inclusiveness: total stock market capitalization/GDP (%); number of listed companies (pcs); and financial institutions' farm savings (billion yuan). Numerosity: length of fiber optic cable lines (km) and number of broadband access subscribers (million). The above indexes are standardized and determined using the entropy method, and the overall evaluation index (FTI) derived from this comprehensive evaluation system range from 0 to 1. The regression results are listed in column (3) of Table 4. The regression coefficients and significance levels for the level of development of digital financial inclusion (FTI) do not vary significantly when the regressions are estimated after the core explanatory variables are replaced.

### 4.3.3. Endogeneity Test

To avoid the endogeneity problem arising from reverse causality and omitted variables, this study attempts to address the estimation bias caused by endogeneity using an instrumental variable method. Drawing on the method described by Zhang Lin (2020) [46], a one-period lag in the digital inclusion index serves as an instrumental variable to maximally eliminate the endogeneity problem triggered with the reverse causality of "the better the development of digital villages, the better the development of digital inclusion". However, there are endogeneity biases (e.g., omitted variables) in the empirical regression equation. Thus, this study draws on the method of Wang Liang (2023) [47] to address the endogeneity issue using rural–urban broadband penetration as an instrumental variable.

Columns (1)–(4) in Table 5 list the regression results of the instrumental variables approach. First, the first-stage regression results are presented in columns (1) and (3). The coefficient for the number of urban and rural broadband households is significant at 1% and has a positive sign, suggesting that the higher the number of urban and rural broadband households, the higher the level of digital inclusive finance. The coefficient for the lagged period of digital inclusive finance is significant at 1% and has a positive sign, suggesting that the higher the level of the lagged period of digital inclusive finance, the higher the level of digital. The coefficient for the first period of digital inclusive finance is significant at 1%. Second, the F-values for the two instrumental variables at the first stage are significantly larger than the critical value of 10, reaching 10.94 and 38.2, respectively. According to the Stock–Yogo judgment criteria, both instrumental variables pass the weak instrumental variable test, i.e., it is appropriate to select urban and rural broadband penetration and digital inclusive finance lagged by one period as variables for the level of digital inclusive finance development. Lastly, the regression results of the second stage are listed in columns (2) and (4), where the level of digital inclusive finance is significant at 1% with a positive sign for the coefficient, suggesting that digital inclusive finance development can indeed

enhance the construction of digital villages, proving the reliability of the regression results in this study.

**Table 5.** Endogeneity test.

| Variables | Urban and Rural Broadband Penetration | | FT-lag1 | |
| --- | --- | --- | --- | --- |
| | Stage 1 | Stage 2 | Stage 1 | Stage 2 |
| FT | 10.287 *** | 2.378 *** | 2.756 *** | 1.186 *** |
| | (1.957) | (0.869) | (0.399) | (0.425) |
| Control | Yes | Yes | Yes | Yes |
| R2 | | 0.842 | | 0.853 |
| Phase I F-value | 10.94 | | 38.2 | |

"***" indicates significance at the 1% level of significance.

### 4.4. Pathway Mechanism Analysis

In the previous study, empirical data were used for an overall and differential analysis of the correlation between the effect of digital inclusive finance development and the construction of the digital village, whereas the effects exerted by the transmission mechanisms involved were not investigated in depth. Thus, in this section, the mediating transmission mechanism is identified and tested from the perspective of science and technology innovation and infrastructure based on the insights of digital inclusive finance development on digital village construction. (1) Science and technology innovation. In the existing research, positive and side-by-side related discussions have been conducted on science and technology innovation as a vital transmission factor. Using science and technology innovation as a mediating variable, Li Linhan (2022) [48] confirmed that digital inclusive finance can significantly contribute to science and technology innovation. This scholar also explained that digital inclusive finance is capable of breaking the barriers of knowledge and technology from the channels, modes, and scope of science and technology spillover, facilitating the multi-proliferation of science and technology from point-to-point to point-to-face, thus increasing the efficiency of industrial innovation, and continuously enhancing industrial competitiveness. Wu Xiaoxi (2021) [49] highlighted that science and technological innovation in the new era affects the development of villagers, villages, and rural industries by enhancing the overall appearance of the countryside, while it can provide technical support to boost the digital construction of the countryside. (2) Infrastructure. Infrastructure can also mediate the transmission of digital inclusive finance for digital villages, a view shared by many scholars. Ma Hongmei (2022) [50] suggested that the regionally coordinated development of digital inclusive finance can be conducive to the construction of information infrastructure, and adjusting the subsidies for SMEs and technology-based enterprises upward can lead to a lower cost of infrastructure construction for enterprises. Moreover, Zhao Xinyu (2022) [51] reported infrastructure development as a vital opportunity for the development of the countryside, and Internet big data platforms take on a certain significance in rural governance and development. (3) Digital literacy. Digital literacy takes on critical significance as a mediating transmitter in digital inclusive finance for constructing digital villages; this view has been supported by numerous scholars. Xie Juan (2022) [52] suggested that digital inclusive finance is capable of effectively enhancing the digital capability and financial literacy of people who have been lifted out of poverty, enhancing their ability to adapt to the publicity, marketing, and business capabilities of the digital economy, and accumulating capital for farmers in areas lifted out of poverty to share the benefits of the digital economy and move towards common services. Wu Xiaolong (2023) [53] highlighted that digital literacy can significantly stimulate farmers' participation in rural digital governance. In the field of public services, an improvement in farmers' digital literacy contributes to the digital transformation of public services in rural areas.

Regarding the regression results in Table 6, column (1) lists the major effect of digital inclusive finance development on digital village construction, columns (2) and (3) present

the tests for the mediating effect of technology creation, columns (4) and (5) are tests for the mediating effect of infrastructure, and columns (6) and (7) represent tests for the mediating effect of education level. As indicated by the main effect, the coefficient for digital inclusive financial development is positive at 1%, suggesting that digital inclusive financial development can significantly contribute to the construction of digital villages. The regression results for the mediating effect of technology innovation suggest that the coefficients in columns (2) and (3) are positive and significant at 1%, with the mediating effect of technology innovation reaching 40.79% (140.985 × 0.003/1.037), and the direct effect of digital inclusive finance reaching 58.92% (0.611/1.037). The direct effect of digital inclusive finance reaches 58.92% (0.6/1.037), suggesting that digital inclusive finance can lay a solid basis for technological innovation and increase the efficiency of the circulation of all factors in the economic system while contributing to technological innovation. Following the development of digital inclusive finance, the momentum of regional technological innovation can be stimulated, low-cost and more convenient financial services can be offered for agriculture-associated enterprises and long-tail people, and the rapid spread of the "spirit of science and innovation" can be facilitated. The development of digital villages was stimulated with the development of digital inclusive finance. As revealed by the regression results for the mediating effect of infrastructure, the coefficients in columns (4) and (5) are positive and significant at 1%, with the mediating effect of infrastructure reaching 19.63% (50.883 × 0.004/1.037), whereas the direct effect of digital inclusive finance is 82.16% (0.852/1.037). As indicated by the above results, the development of digital inclusive finance is driving the trend of a "new infrastructure wave" in companies and third-party organizations in mobile communication infrastructure, such that the construction of infrastructure has been significantly facilitated. Moreover, the wave of mobile communication infrastructure construction brought about with digital inclusive finance expedited an improvement in the modern face of the countryside, such that a new chapter has been opened in the construction of the digital countryside. Furthermore, the wave of mobile communication infrastructure brought about with digital inclusion has boosted an improvement in the modernization of the countryside while opening a new chapter in the construction of the digital village. As revealed by the regression results for the mediating effect of digital literacy, the coefficients in columns (6) and (7) are positive and significant at 1% and 10%, with the mediating effect of education level reaching 15.37 (19.922 × 0.008/1.037), whereas the direct effect of digital inclusive finance is 84.09 (0.872/1.037). The development of digital inclusive finance can lay a favorable basis for enhancing the digital literacy of the regional population. Long-tail people or agricultural-associated enterprises will take the initiative to improve their digital literacy to obtain financial support, and residents with higher digital literacy play an intuitive role in boosting the construction of a digital village.

**Table 6.** Regression results of the intermediary mechanism test.

| Variables | Digital Village (1) Stage 1 | Science and Technology Innovation (2) Stage 2 | Digital Village (3) Stage 3 | Communication Infrastructure (4) Stage 2 | Digital Village (5) Stage 3 | Education Level (6) Stage 2 | Digital Village (7) Stage 3 |
|---|---|---|---|---|---|---|---|
| FT | 1.037 *** (0.242) | 140.985 *** (36.988) | 0.611 *** (0.223) | 50.883 *** (17.306) | 0.852 *** (0.240) | 19.922 *** (3.577) | 0.872 *** (0.259) |
| TI | | | 0.003 *** (0.001) | | | | |
| MCI | | | | | 0.004 *** (0.001) | | |
| DL | | | | | | | 0.008 * (0.005) |
| Control | Yes | Yes | Yes | Yes | Yes | Yes | Yes |
| Constant | −7.623 *** (0.676) | −851.076 *** (103.160) | −5.050 *** (0.697) | −10.614 *** (48.266) | −7.584 *** (0.654) | −44.470 *** (9.977) | −7.254 *** (0.706) |
| Fixed time | Yes | Yes | Yes | Yes | Yes | Yes | Yes |
| Fixed area | Yes | Yes | Yes | Yes | Yes | Yes | Yes |
| R2 | 0.974 | 0.930 | 0.980 | 0.943 | 0.976 | 0.993 | 0.975 |

"***" and "*" indicate significance at the 1% and 10% levels of significance.

In brief, there is indeed a partial mediating effect based on science and technology innovation, infrastructure construction, and digital literacy in the process of digital inclusive financial development contributing to the construction of digital villages. In other words, with the development of digital inclusive finance, the construction of digital villages can be directly promoted, whereas the construction of digital villages can be boosted by stimulating the development of technological innovation, infrastructure construction, and digital literacy level. As revealed by the above result, Hypothesis 3 of this study is valid.

### 4.5. Heterogeneity Analysis

4.5.1. Inter-Territorial Heterogeneity

The sample data fall into regions for step-by-step analysis, with the aim of investigating the heterogeneity in the effect of digital inclusive finance development on the construction of digital villages. Table 7 lists the results of the regional heterogeneity in the analysis of the effect of digital inclusive finance development on digital village construction by dividing the sample data into eastern, central, and western regions. In general, the regression results for the eastern and western samples suggest that digital inclusive finance significantly contributes to the construction of the digital village, while the central region does not report any significant effect. The degree of the effect differs between the east and west, with the maximum impact coefficient (2.075) in the east, where the level of infrastructure and economic development is higher, increasing the efficiency of digital village construction by 2.075 for every 1% increase in the level of digital inclusive financial progress. The second maximum impact coefficient (0.646) is in the west, increasing the efficiency of digital village construction by 0.646 for every 1% increase in the level of digital inclusive finance. The possible reasons for the above result are the more economically developed eastern region, a higher standard of living of rural residents, and the strong financial needs of enterprises and residents. Although, the progressive development of digital inclusive finance can increase financial accessibility and the efficiency of the productive lives of enterprises and resident groups, such that they yearn for the convenience brought about with the construction of the digital village. As a result, the promotion of the construction of the digital village in the eastern region can be boosted. The disadvantage in the economic development of the western region turns out to be significant, rural residents raise an urgent need for financial services, and the marginal effect of residents' access to financial services is much higher than that in the eastern region, such that digital inclusive finance is promoted to enhance the efficiency of the construction of digital countryside. The central region, on the other hand, is sandwiched in the middle. Despite its large population and certain digital base, its overall economic level is relatively low, its digital infrastructure needs to be updated, and its residents are less financially active, requiring the government to provide substantial financial support for improving the rural landscape in a holistic manner. Here, the development of digital inclusive finance cannot directly stimulate the construction of the digital countryside. Thus, the geographically differentiated effect of digital inclusive finance in H2 is verified, suggesting the strong robustness of the core findings of this study.

**Table 7.** Empirical results of the regional differential effect of digital inclusive finance on digital village construction.

| Variables | (1) East | (2) Middle | (3) West |
|---|---|---|---|
| FT | 2.075 *** | 0.138 | 0.646 ** |
|  | (0.554) | (0.465) | (0.286) |
| Control | Yes | Yes | Yes |
| Constant | −5.014 *** | −0.002 | −0.181 |
|  | (1.012) | (0.089) | (0.0745) |
| Fixed time | Yes | Yes | Yes |
| Fixed area | Yes | Yes | Yes |
| R2 | 0.986 | 0.930 | 0.971 |

"***" and "**" indicate significance at the 1% and 5% levels of significance.

#### 4.5.2. Inter-Dimensional Heterogeneity

Digital inclusive finance exhibits diverse characteristics, and a reflection on the effect of multi-dimensional indexes on digital villages should be covered in an analysis of the effect of digital inclusive finance development on the construction of digital villages, with the aim of drawing normative conclusions. As depicted in Table 8, the overall regression results for the degree of coverage do not take on significance, whereas the depth of use and the degree of digitalization are significant at 1%. For the degree of impact, the regression coefficient for depth of use is 0.601, having the largest effect, whereas the regression coefficient for digitization reaches 0.361, having the second largest effect. From the perspective of the impact on the construction of digital villages, first, the breadth of coverage index of digital inclusive finance refers to the measurement of the proportion of Alipay card-tied users, the number of accounts, and the number of card-tied users, which mostly indicates the audience of a single group. However, the current financial-associated supporting facilities in China are immature and the digital quality of residents is relatively low, such that the effect on the construction of digital villages is slight. Second, the use of depth indexes expresses the services (e.g., credit loans and commercial insurance provided with digital inclusive finance) to alleviate the financing constraints of agriculture-associated MSMEs, providing financing convenience, asset allocation, and risk control for MSMEs, thus providing effective assistance to the construction of digital villages. Third, the effect of digitalization is lower than the depth of use. The underlying reason continues to be the inadequate construction of China's information digitalization network and relevant supporting infrastructure, and the degree of digitalization requires long-term development to make a qualitative breakthrough. Thus, although the effect on the construction of the digital village is significant, the degree of effect is lower than the depth of use. Accordingly, the dimensional difference effect of digital financial inclusion in H2 is verified, thus proving the robustness of the core conclusion of this study.

**Table 8.** Empirical results of the differential effect of digital financial inclusion on the dimensionality of digital village construction.

| Variables | (2) Digital Village | (4) Digital Village | (6) Digital Village |
|---|---|---|---|
| FT1 coverage breadth | −0.156 (0.331) | | |
| FT2 depth of use | | 0.601 *** (0.133) | |
| FT3 digitization | | | 0.361 *** (0.083) |
| Control | Yes | Yes | Yes |
| Constant | −8.017 *** (0.709) | −7.861 *** (0.668) | −7.726 *** (0.673) |
| Fixed time | Yes | Yes | Yes |
| Fixed area | Yes | Yes | Yes |
| R2 | 0.972 | 0.975 | 0.974 |

"***" indicates significance at the 1% level of significance.

#### 4.6. Exploring the Internal Foundations and External Constraints on the Effectiveness of Digital Inclusive Finance

As indicated by the analysis in this study, there is an internal base effect of rural residents' consumption and an external constraint effect of financial regulation on the effectiveness of digital financial inclusion.

On the one hand, the moderating role played by the internal basis of rural residential consumption is investigated. Most scholars suggested that the growth of resident consumption can positively facilitate digital inclusive financial residence [54], and considerable consumption stimulation can catalyze the diversity of financial derivation and the formation of financial products with the characteristics of the target clientele. Thus,

encouraging rural residents' consumption awareness has positive practical implications for promoting the development of digital inclusive finance and further building a digital village. Accordingly, this study uses the consumption number of rural residents in a respective province to examine the consumption level of rural residents and empirically studies the influence status of rural residents' consumption in the process of using digital inclusive finance to expedite the digital village. In Table 9, column (1) shows the regression results of the cross-section of digital inclusive finance indexes and rural resident consumption on the digital village. The coefficient for the cross-section is positive and significant at 5%, which means that rural residents' consumption drives the positive effect of digital inclusive finance on the digital village.

**Table 9.** The moderating effect of residential consumption and financial regulation.

| Variables | Digital Village (1) | Digital Village (2) |
|---|---|---|
| Digital inclusive finance | 0.2417 * (0.147) | 0.650 *** (0.086) |
| Rural consumer | 0.892 *** (0.300) | |
| Financial regulation | | 4.580 ** (2.159) |
| Digital inclusive finance × consumption by rural residents | 0.457 ** (0.203) | |
| Digital inclusive finance × financial regulation | | 81.921 ** (33.154) |
| Control variables | Control | Control |
| Constant term | 0.005 *** (0.002) | −0.003 *** (0.023) |
| Time fixed | Yes | Yes |
| Region fixed | Yes | Yes |

"***", "**" and "*" indicate significance at the 1%, 5% and 10% levels of significance.

On the other hand, the regulatory role of external constraints on financial regulation is investigated. Tang Song (2020) [55] argue that digital financial inclusion has not changed the "risk-reward" principle of the financial industry, but rather the digital features will allow financial risks to quickly permeate the entire financial system [56]. As a result, stronger financial regulation has become necessary. Most scholars agree that financial regulation can effectively reduce the probability of arbitrage, liquidity risk and real financialization in the financial sector, and ensure the standardized and safe operation of financial markets, thus providing better credit and fund services to farmers and increasing social trust. Accordingly, this study uses government financial regulatory expenditure as a proportion of the value added to the financial sector to examine the strength of unified financial regulation and empirically explores the potential effect of financial regulation at this stage in the process of using digital financial inclusion to expedite the digital village. In Table 9, column (2) indicates the regression results for the cross-section of digital inclusive finance indexes and financial regulation on the digital village. The coefficient for the cross-section is positive and significant at 5%, suggesting that financial regulation promotes the positive effect of digital inclusive finance on the revitalization of the village.

## 5. Conclusions and Recommendations

### 5.1. Conclusions

In this study, the mechanism underlying the role of digital inclusive finance development in promoting the construction of the digital countryside is first theoretically classified.

Subsequently, based on the digital inclusive finance index published by the Digital Research Centre of Peking University 2020 and the macro statistics on 30 Chinese provinces and municipalities from 2013 to 2020, the digital inclusive finance mechanism of action and mediated transmission mechanism underlying development are empirically investigated using a two-way fixed panel model and a mediating effect model to expedite the construction of digital villages. Finally, the geographical and dimensional heterogeneity is tested.

As indicated by the result of this study: first, the development of digital inclusive finance can positively and significantly boost the construction of digital villages and has become an important driving force in the construction of a modern Chinese style village in the new era. This conclusion holds after stability tests (e.g., replacing core variable measures, excluding random and developed provincial and municipal samples, and instrumental variable methods). Second, heterogeneity exists in the role of digital inclusive financial development in boosting the construction of the digital countryside, with eastern provinces and cities having a stronger role in promoting the construction of the digital countryside compared with central and western regions. Thus, the unevenness in the current development of digital inclusive finance in promoting the construction of digital villages is revealed. Third, digital inclusive finance can indirectly expedite the implementation of digital village construction with the positive mediating effect of promoting technological innovation and communication infrastructure construction, suggesting that technological innovation, communication infrastructure construction, and residents' education level are vital paths for digital village construction under Chinese-style modernization in the new era. Fourth, a positive moderating effect of internal conditions of rural resident consumption and external conditions of financial regulation is reported when digital inclusive finance promotes digital rural development.

*5.2. Policy Recommendations*

Given the above findings, the following policy recommendations are made:

First, following the aim of building Chinese modernization, China should implement differentiated policies based on regional resource endowments, social conditions, and rural characteristics and tailor them to local conditions. China should support the promotion of digital inclusive finance in the central and western provinces, bridge the digital financial gap between different regions, and eliminate financial exclusion by conforming to several complementary policies, mainly on inclusive finance for the "three rural areas", together with tax concessions, financial subsidies, and the establishment of a negative list. It is imperative for the developed provinces in the eastern region to develop a large-scale digital network for exploiting the spatial spillover effect and spreading the positive effect of digital inclusive finance to the central and western regions, such that a "fast-led" model of support can be formed. China should also give full play to the value of data as a factor for production, facilitate the open sharing of data in an orderly fashion, boost the integration of data between government and enterprises, enterprises and enterprises, and enterprises and individuals, and drive the healthy development of the digital countryside based on digital inclusive finance.

Second, China should vigorously expedite the digitalization process in rural areas, optimize and upgrade mobile communication infrastructure, and inject new momentum into the construction of its digital countryside. With the continuous promotion of China's "Digital China" strategy, mobile Internet infrastructure turns out to be the largest worldwide, whereas the status quo of "more but not stronger, bigger but not better" persists. During the "14th Five-Year Plan" period, major infrastructure projects and works are urgently needed to expand and upgrade fiber-optic networks, cover special environments with 5G base stations, build interactive artificial intelligence facilities, and expedite spectrum-based blockchain technology. Furthermore, the integration of digital technology and rural construction should be expanded to optimize the digital governance system of the countryside

and fully release digital dividends to provide rural residents with more convenient access to financial services.

Third, the investment in scientific and technological research and development and high-level innovation talents should be increased to enhance the independent innovation capacity of digital technology and facilitate the upgrading of innovation. Seizing the historical opportunity of Chinese modernization, the construction of a science and technology innovation system can be facilitated, and the strengths of the three parties can be integrated, i.e., industry, academia, and research, to place a focus on breaking through the "neck" of key digital technologies. China should also increase the popularity and promotion of the "government-led, private investment" model, so that more capital through multiple channels converge on the track of scientific and technological development and talent innovation, for the incubation of hard science and technology innovative talent teams. The cultivation of digital financial inclusion industry clusters can be accelerated, the layout of innovative industrial chains for digital financial technology can be facilitated, and the application of digital financial technology can be deepened in more financial scenarios. Next, data elements and digital technology can be further used to lead technological innovation, the effect of digital inclusive finance and technological innovation in the construction of the digital village can be activated, and the construction of digital China and the digital village can be fully empowered.

Fourth, the investment in education should be increased to enhance the digital literacy of urban and rural residents. Targeted training programs should be developed, and farmers' awareness of digital life, prevention, and privacy protection in the countryside can be strengthened using video lectures, door-to-door distribution of brochures, special lectures, and the presentation of typical cases. China should continue to enrich new digital application scenarios and deepen farmers' digital participation in five major scenarios, including digital economy, digital ecology, digital culture, digital livelihood, and digital governance, and continue to empower the cultivation of farmers' digital literacy by continuously widening the coverage of application scenarios, so as to enhance the service experience of farmers' digital production and life in different application scenarios.

Fifth, the enthusiasm of rural residents should be actively promoted to consume and achieve a healthy balance in financial regulation. The development of digital inclusive finance has led to a constant turnover of financial products, such that rural residents are stimulated to expand their consumption with the diversity and convenience of financial products, and a positive dynamic is created toward the digital village. Moreover, the rigid requirements of financial regulation should not be relaxed to curb conditions (e.g., non-performing loans and information fraud), such that a virtuous balance can be created between financial regulation and the coordination of the digital countryside.

*5.3. Further Discussion*

The development of digital inclusive finance takes on profound significance in the construction of digital villages, and this study explores a range of these issues and uses empirical evidence for verification. Although the discussion takes on theoretical and practical significance, there are still some limitations in this study, which are expected to be remedied in future studies. On the one hand, certain shortcomings exist in the research sample. First, the sample used in this study represents the provincial level in China. To study the issue of digital villages in depth, data can be further selected from the city and county level to be more representative. Second, the sample used lacks an international perspective. Data from developed countries (e.g., the United States, the United Kingdom, and Italy) and developing countries (e.g., India and Malaysia) should be considered for discussion and validation, with the aim of further indicating the development of the global digital village. On the other hand, in this study, the stress is placed on the effect of digital inclusive finance on the construction of the digital village. With the aim of investigating the issues regarding the digital village in depth, we can delve into the issues of digital

governance of the village and the digitization of rural industries and study more deeply the various aspects of the construction of the digital village.

**Author Contributions:** Conceptualization, C.Z. and Y.L.; methodology, C.Z.; software, C.Z.; validation, C.Z. and Y.L.; formal analysis, C.Z. and Y.L.; investigation, C.Z. and Y.L.; resources, C.Z., Y.L., L.Y. and Z.W.; data curation, C.Z.; writing—original draft preparation, C.Z. and Y.L.; writing—review and editing, Z.W., Y.L. and L.Y.; visualization, C.Z. and Y.L.; supervision, Z.W.; project administration, Z.W.; funding acquisition, C.Z. and Z.W. All authors have read and agreed to the published version of the manuscript.

**Funding:** This study was funded by the Beijing Academy of Science and Technology municipal financial project "Research on the coordinated development of Beijing-Tianjin-Hebei digital economy" (1420238669KF001-02) and "Research on Application Scenario-Driven Market Allocation of Data Elements in Beijing" (1420238669KF001-03).

**Institutional Review Board Statement:** Not applicable.

**Data Availability Statement:** The data used to support the findings of this study are available from the corresponding author upon request.

**Conflicts of Interest:** The authors declare no conflict of interest.

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
