# Peer review of "Does the Development of Digital Inclusive Finance Promote the Construction of Digital Villages?—An Empirical Study Based on the Chinese Experience"

_agriculture, doi:10.3390/agriculture13081616_

Round 1

Reviewer 1 Report

I have reviewed your paper and have some constructive comments to help you improve it:

Citations: There are several issues with the citations throughout the paper, such as "(Li Heng English, 2023) [1]" (line 38). These errors are recurrent on multiple pages. I recommend thoroughly reviewing the citation format and ensuring consistency and accuracy throughout the paper.

Introduction: The introduction section has some issues that need to be addressed. Starting from line 65, you refer to a review of existing research, but it is missing from the paper. To justify your paper's objectives based on existing research, ensure you provide the relevant evidence and adjust the sentence accordingly.

Additionally, the study's objectives are somewhat confusing. It would be better to consolidate the objectives into one or two main ideas. For example, from line 75, you mention objectives related to exploring heterogeneity based on different samples and investigating the role of special factors in the effect of digital inclusive finance on the construction of digital villages. These objectives add complexity to your study. I suggest providing a clear, concise, and objective statement about the paper's objectives  (one or two) to avoid confusion and highlight the main breakthrough your study aims to achieve.

Moreover, in the last paragraph of the introduction section, you use Roman numerals, while the paper's headings are organized in Arabic numerals. It would be better to standardize this issue for consistency.

Literature Review: The literature review section is interesting, but the organization of ideas needs improvement. The sentences are somewhat confusing, and Figure 1 does not bring clarity to the proposed model. I recommend providing a clearer argument for the hypothesis statements and using a simpler figure that portrays the study's hypotheses more effectively. Consider using arrows to represent each hypothesis in Figure 1.

Research Design: You should provide a clear and simple justification for choosing the variables used in the study. While you make some arguments, a more robust justification for why these specific measures were chosen is necessary. Additionally, using outdated data might be problematic, considering the rapid development of technology and significant changes in world geopolitics, including the impact of the COVID-19 pandemic. If you believe that these changes do not affect the study's objectives, provide a clear explanation. If more recent data is unavailable, state the reasons for using older data.

Furthermore, offer a better justification for the methods used in the study, such as why an OLS model was employed and the software used for analysis. Also, mention the date when the data was retrieved to provide readers with context.

Conclusions and Recommendations: The conclusions and recommendations provide valuable insights, but improvements are needed in the writing style. For instance, starting from line 784, the phrase "Second, we will vigorously expedite the digitalization process in rural areas..." raises questions about who "we" refers to. Specify the source or target audience of this statement, especially if it is an official report from policymakers.

I hope these suggestions help you improve your paper.

Good luck with your work!

Author Response

Cover letter

        We are very grateful for your professional review of our article. As you pointed out, there are several issues that need to be addressed. Following your kind suggestions, we have made extensive corrections to our previous draft, the detailed corrections are listed below.

        In the quotes section. We have made changes to the issues in "(Li Heng English, 2023) [1]", removed incorrect vocabulary, hidden the author's name, and conducted a comprehensive review of the entire literature to ensure accurate correspondence in each document.

        In the introduction. Firstly, the sentence in the literature review that you mention on line 65 was an error in our writing. Detailed literature review content should not be included in this section, as there will be detailed literature reviews later. We have therefore deleted and amended the original sentence structure to ensure consistency throughout the text. Secondly, the issue of confusion in the research aims that you raise has also been revised. We have consolidated several objectives into two main points, thereby achieving clarity and consistency in the research objectives. Thirdly, regarding the issue of the Roman character in the last paragraph, we have also made adjustments to achieve consistency throughout the text.

        In the literature review section. We have added a distinction between the theoretical and practical parts, and further elaborated on the historical evolution of digital inclusive finance in financial development theory in development economics. And further improved the language description of the literature review, adding the hypothesis of regulatory effects related to the mechanism diagram. In addition, we redesigned and optimised the images according to the mechanism diagram issue you mentioned, so that readers can read them more intuitively.

        In the Research and Design section. We had a heated discussion about the questions you raised, "Why is this research data?" and "COVID-19 pandemic". We will provide you with a formal response to this. The data we selected comes from internal government data, some statistical yearbooks and the Digital Inclusive Finance Index published by Alibaba Research Institute. These data are complete and can be measured from 2013 to 2020, but some data after 2021 have not been released and cannot be included in the analysis. Therefore, we chose to conduct econometric calculations and analysis from 2013 to 2020. During this period, China's digital inclusive finance and digital rural construction entered a stage of significant development, with typical fluctuation characteristics and high research value (see our article on sustainability published in May 2023 for this research result).Finally, we provided a detailed explanation of why the OLS model was used, what software was used, and the date of data retrieval.

        In the conclusion and suggestions section. We have made revisions to the wording and subject matter, resulting in a clearer overall style.

        We hope that our answers cover the issues you raised and that the revised content will further improve the quality of the article. Thanks again for your feedback on our article! Wish you all the best!

Reviewer 2 Report

I would like to thank the authors for the interesting and current research topic. I would like to make some suggestions in order to improve the quality of the manuscript, which does not diminish its importance.

The abstract is fully adequate and satisfactory. The authors have provided extensive information in the introduction as well as in the literature review.

The hypotheses are clearly stated and the methodology has no objections. The research results are presented clearly and very comprehensibly. The figures are very clear. No corrections are needed in the rest of the manuscript.

1. What is the main question addressed by the research?

The main issue addressed in this manuscript relates to the impact of the development of digital inclusive finance on the promotion of the construction of digital villages. A study based on the Chinese experience is presented. One of the objections was just to change the title, so that it would not be in the form of a question.

2. Do you consider the topic original or relevant in the field? Does it
address a specific gap in the field?

I am of the opinion that the topic is relevant in this field and that it can help supplement the scarce literature in that field, but also have practical implications.

3. What does it add to the subject area compared with other published
material?

The subject area of research provides innovations in terms of research methods, supplementing the insufficient number of researches on this issue, but also introduces innovations when it comes to the application of methodology in this type of research and in this specific area.

4. What specific improvements should the authors consider regarding the
methodology? What further controls should be considered?

I am of the opinion that the authors applied an adequate methodology, which they primarily explained very well to the readers. No changes are necessary in the part related to research methodology.

5. Are the conclusions consistent with the evidence and arguments presented
and do they address the main question posed?

Concluding considerations, as well as results, are given very clearly and explained in detail. Each segment of the research is presented clearly and precisely. The obtained results were compared with previous research, and some suggestions were given for the future implications of the results and future similar research and application of the results.

6. Are the references appropriate?
The references are adequate and I suggest that the authors check the way of citing references in the literature review section. I think it is necessary to keep only the numbers in parentheses, and delete the names of the authors in the parentheses next to them.
I suggest supplementing the references: Stereotypes and Prejudices as (Non) Attractors for Willingness to Revisit Tourist-Spatial Hotspots in Serbia. http://dx.doi.org/10.3390/su15065130

7. Please include any additional comments on the tables and figures.

All figures and tables are clear, with a very good resolution displayed and give all the necessary results in a very simple and clear way. No changes are required.

Author Response

Cover letter

      We greatly appreciate your professional review of our article. As you pointed out, there are several issues that require attention. Based on your helpful suggestions, we have made significant revisions to our prior draft. Please find the detailed corrections listed below.

     We have addressed the concern you raised in the references by making the necessary modifications and removing all names in parentheses. This ensures consistency and accuracy throughout the literature. Furthermore, we have diligently reviewed the literature you suggested and unanimously concluded its helpfulness to our research. We have cited this literature in our paper (42).

     We trust that our responses address the questions you raised and the revised content enhances the overall quality of the article. Thank you once again for providing feedback on our article! Wish you all the best! 

Reviewer 3 Report

Dear authors,

I would like first of all to congratulate you on your work. I found it very interesting and a great job has been done on the methodological part. Although the sample is 13-20, I do not consider it necessary to increase the time period since the fundamentals of the article maintain its relevance.

My biggest concern is related to the motivation on which the hypotheses rest. Although I find that the hypotheses could make sense, they are justified in general comments like what academics find, or it is known in the literature, etc. I consider that it is not enough. It is the only part of the work that I have found weak and therefore I recommend reinforcing.

I found the most interesting part in the discussion of how to implement differentiated policies, although I would broaden the discussion with literature on what actions can be taken to vigorously promote these infrastructures.

If possible, I would like to broaden the discussion between the relationship between scientific and technological research and development funds and high-level innovative talent and the innovation variable, proposing at least some referenced measure.

Lastly, I would like more detail on how the digital literacy variable has been constructed. It seems that it is a proxy for the number of professors but I consider it necessary to extend the explanation to the reader.

I hope you find the comments useful.

Author Response

Cover letter

        We greatly appreciate your professional review of our article. As you pointed out, there are several issues that require attention. We made extensive corrections to our previous draft based on your helpful suggestions. The detailed changes are listed below.

        We have adjusted the motivation behind the hypothesis that you mentioned. We have examined and organized the theory of financial development in the economic development of digital inclusive finance, and explained it in both theoretical and practical terms. This has enhanced the foundation of the hypothesis and increased the overall academic significance of the article.

        We have elaborated in detail on the suggestion you made about "infrastructure" in the second part, and we feel it should correspond to your comprehension of this aspect.

       We have not extensively discussed the correlation between research and development funds, highly skilled innovators, and innovation factors. So, we have added more information to the third part of the suggestion, with the intention of improving this discussion.

        We've explained digital literacy variables in greater detail in the variable explanation section, to clarify how they're constructed and facilitate better understanding for our readers.

       We trust that our responses have addressed your concerns, and the revised content can enhance the quality of the article. Thank you for giving us your feedback on our article again! Wish you all the best!

Round 2

Reviewer 1 Report

Dear authors,

 I’m satisfied with the changes made in the paper.

Good luck with your work.